# Neural Modeling of Multi-Predicate Interactions for Japanese Predicate Argument Structure Analysis

## Abstract

The accuracy of Japanese predicate argument structure (PAS) analysis has improved in recent years thanks to joint modeling of interactions between multiple predicates. However, this approach heavily relies on syntactic information predicted by parsers, and suffers from the error propagation. To remedy this problem, we introduce a model using *grid-type recurrent neural networks* (Grid-RNN), which automatically induces features sensitive to *multi-predicate interactions* from word sequence information of a sentence. The experiments on the NAIST Text Corpus show that our model exceeds the accuracy of the state-of-the-art Japanese PAS analyzer without syntactic information.

## 1 Introduction

Predicate argument structure (PAS) analysis is a basic semantic analysis task, in which systems are required to identify semantic units of a sentence, such as *who did what to whom*. In pro-drop languages such as Japanese, Chinese and Italian, arguments are often omitted in text, and such *argument omission* is regarded as one of the most problematic issues of PAS analysis (Iida and Poesio, 2011; Sasano and Kurohashi, 2011).

As an approach to the argument omission problem, in Japanese PAS analysis, joint modeling of interactions between multiple predicates has been gaining popularity and achieved the state-of-the-art result (Ouchi et al., 2015; Shibata et al., 2016). This approach is based on the linguistic intuition that the predicates in a sentence are semantically related to each other and the interaction information can be a clue for PAS analysis. However, to model such *multi-predicate interactions*, this ap-

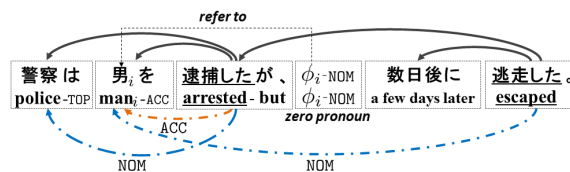

( The police arrested the man, but $\phi$ escaped a few days later )

Figure 1: An example of Japanese PAS. The upper edges denote dependency relations, and the under edges denote case arguments. "NOM" and "ACC" represent the nominative and accusative arguments, respectively. "$\phi_i$" is a *zero pronoun*, referring to the *antecedent* "男 $_i$ (man$_i$)".

proach heavily relies on syntactic information predicted by parsers and suffers from the error propagation caused by the pipeline processing.

To remedy this problem, we propose a neural model which automatically induces features sensitive to multi-predicate interactions from word sequence information of a sentence. This model takes as input all predicates and their argument candidates in a sentence at a time, and captures the interactions using grid-type recurrent neural networks (Grid-RNN) without syntactic information.

In this paper, we firstly introduce a basic model using RNNs, which independently estimates arguments of each predicate without considering the multi-predicate interactions (Sec. 3). Then, extending this model, we propose a neural model using Grid-RNNs (Sec. 4).

Performing experiments on the NAIST Text Corpus (Iida et al., 2007), we demonstrate that our neural models exceed the accuracy of the state-of-the-art Japanese PAS analyzer (Ouchi et al., 2015). In particular, the neural model using Grid-RNNs achieves the best result, which suggests that our grid-type neural architecture effectively captures multi-predicate interactions. [1]

---

[1] Our source code is publicly available at `http:xxx`

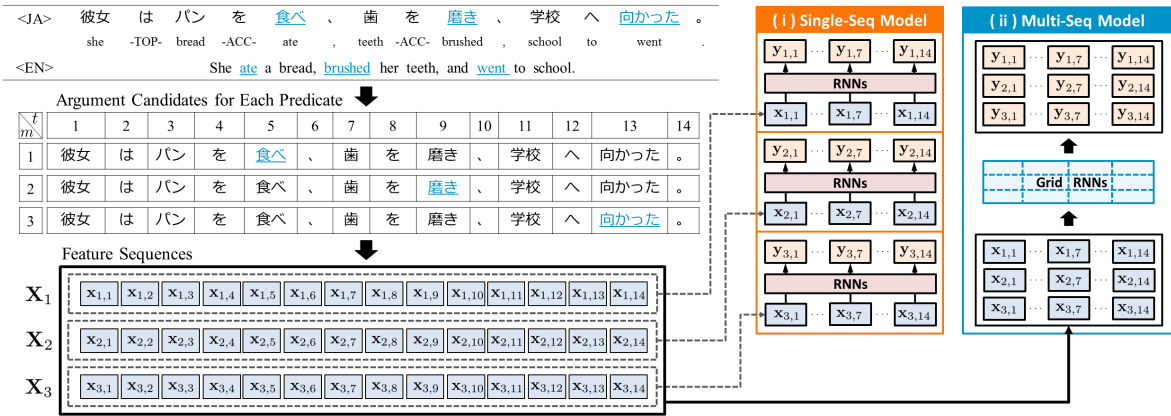

Figure 2: An overview of the neural models: (i) *single-sequence* and (ii) *multi-sequence* models.

## 2  Japanese Predicate Argument Structure Analysis

### 2.1  Task Description

In Japanese PAS analysis, we identify arguments taking part in the three major case roles, *nominative* (NOM), *accusative* (ACC) and *dative* (DAT) cases, for each predicate. Arguments can be divided into the following three categories according to the positions relative to their predicates (Hayashibe et al., 2011; Ouchi et al., 2015):

*Dep*: The arguments that have direct syntactic dependency with the predicate.

*Zero*: The arguments referred to by zero pronouns within the same sentence, which have no direct syntactic dependency with the predicate.

*Inter-Zero*: The arguments referred to by zero pronouns out of the same sentence.

For example, in Figure 1, the nominative argument "警察 (police)" for the predicate "逮捕した (arrested)" is regarded as a *Dep* argument since the argument has a direct syntactic dependency with the predicate. In contrast, the nominative argument "男 $_i$ (man$_i$)" for the predicate "逃走した (escaped)" is regarded as a *Zero* argument since the argument has no direct syntactic dependency with the predicate.

In this paper, we focus on the analysis for these intra-sentential arguments, i.e., *Dep* and *Zero*. In order to identify inter-sentential arguments (*Inter-Zero*), it is required to search a much broader space, such as the whole document, resulting in a much harder analysis than intra-sentential arguments.[2] Thus, Ouchi et al. (2015)

and Shibata et al. (2016) focused on only intra-sentential argument analysis. Following this trend, we focus on intra-sentential argument analysis.

### 2.2  Challenging Problem

Arguments are often omitted in Japanese sentences. In Figure 1, $\phi_i$ represents the omitted argument, called *zero pronoun*. This zero pronoun $\phi_i$ refers to "男 $_i$ (man$_i$)". In Japanese PAS analysis, when an argument of the target predicate is omitted, we have to identify the antecedent of the omitted argument (*Zero* argument).

The analysis for such *Zero* arguments is much more difficult than that for *Dep* arguments because of the lack of direct syntactic dependencies. For *Dep* arguments, the syntactic dependency between an argument and its predicate is a strong clue. In the sentence in Figure 1, for the predicate "逮捕した (arrested)", the nominative argument is "警察 (police)". This argument can easily be identified by relying on the syntactic dependency. In contrast, since the nominative argument "男 $_i$ (man$_i$)" has no syntactic dependency with its predicate "逃走した (escaped)", we have to use other information for such zero argument identification.

As an solution to this problem, we exploit two kinds of information: (i) context in the entire sentence and (ii) multi-predicate interactions. For the former, we introduce *single-sequence model*, which induces context-sensitive representations from a sequence of argument candidates of a predicate. For the latter, we introduce *multi-sequence model*, which induces predicate-sensitive representations from multiple sequences of argument candidates of all predicates in a sentence (shown in Figure 2).

---

[2]The F-measure remains 10-20% (Taira et al., 2008; Imamura et al., 2009; Sasano and Kurohashi, 2011).

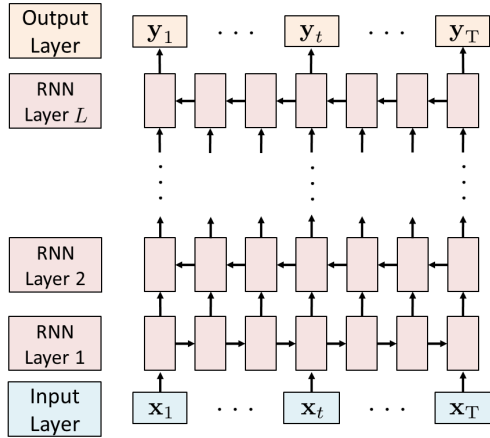

Figure 3: The overall architecture of the single sequence model. This model consists of three components: (i) Input Layer, (ii) RNN Layer and (iii) Output Layer.

## 3 Single-Sequence Model

The single-sequence model exploits stacked bidirectional RNNs (Bi-RNN) (Schuster and Paliwal, 1997; Graves et al., 2005, 2013; Zhou and Xu, 2015). Figure 3 shows the overall architecture, which consists of the following three components:

**Input Layer:** Map each word to a feature vector representation.

**RNN Layer:** Produce high-level feature vectors using Bi-RNNs.

**Output Layer:** Compute the probability of each case label for each word using the softmax function.

In the following subsections, we describe each of them in more detail.

### 3.1 Input Layer

Given an input sentence $w_{1:T} = (w_1, \cdots, w_T)$ and a predicate $p$, each word $w_t$ is mapped to a feature representation $\mathbf{x}_t$, which is the concatenation ($\oplus$) of three types of vectors:

$$\mathbf{x}_t = \mathbf{x}_t^{arg} \oplus \mathbf{x}_t^{pred} \oplus \mathbf{x}_t^{mark} \qquad (1)$$

where each vector is based on the following atomic features inspired by Zhou and Xu (2015):

ARG: Word index of each word.

PRED: Word index of the target predicate and words around the predicate.

MARK: Binary index that represents whether the word is the predicate or not.

| | ARG | PRED | MARK |
|---|---|---|---|
| <JA> | 彼女 は パン を 食べた 。 | | |
| | she -TOP- bread -ACC- ate . | | |
| <EN> | She ate a bread. | | |

Features

| | ARG | PRED | MARK |
|---|---|---|---|
| 1 | 彼女 | を 食べた 。 | 0 |
| 2 | は | を 食べた 。 | 0 |
| 3 | パン | を 食べた 。 | 0 |
| 4 | を | を 食べた 。 | 0 |
| 5 | 食べた | を 食べた 。 | 1 |
| 6 | 。 | を 食べた 。 | 0 |

Figure 4: An example of the feature extraction. The underlined word is the target predicate. From the sentence "彼女はパンを食べた。(She ate a bread.)", the three types of features are extracted for the target predicate "食べた (ate)".

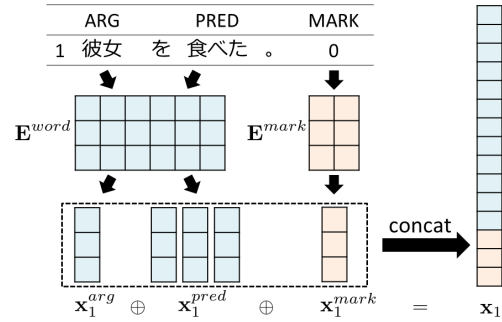

Figure 5: An example of the process of creating a feature vector. The extracted features are mapped to each vector, and all the vectors are concatenated into one feature vector.

Figure 4 presents an example of the atomic features. As the ARG feature, we extract a word index $x^{word} \in \mathcal{V}$ of each word. Similarly, as the PRED feature, we extract each word index $x^{word}$ of the $C$ words taking the target predicate at the center, where $C$ is the window size. The MARK feature $x^{mark} \in \{0, 1\}$ is a binary value that represents whether the word is the predicate or not.

Then, using feature indices, we extract feature vector representations from each embedding matrix. Figure 5 shows the process of creating the feature vector $\mathbf{x}_1$ for the word $w_1$ "彼女 (she)". We set two embedding matrices: (i) word embedding matrix $\mathbf{E}^{word} \in \mathbb{R}^{d_{word} \times |\mathcal{V}|}$, and (ii) mark embedding matrix $\mathbf{E}^{mark} \in \mathbb{R}^{d_{mark} \times 2}$. From each embedding matrix, we extract corresponding column vectors and concatenate them as a feature vector representation $\mathbf{x}_t$ based on Eq. 1.

Each feature vector $\mathbf{x}_t$ is multiplied with a parameter matrix $\mathbf{W}_x$:

$$\mathbf{h}_t^{(0)} = \mathbf{W}_x \, \mathbf{x}_t \qquad (2)$$

The vector $\mathbf{h}_t^{(0)}$ is given to the first layer of the RNN layers as input.

### 3.2 RNN Layer

In the RNN layers, feature vectors are updated recurrently using Bi-RNNs. Bi-RNNs process an input sequence from the left-to-right manner in odd-numbered layers and the opposite in even-numbered layers. By stacking these layers, we can construct the deeper network structures.

Stacked Bi-RNNs consist of $L$ layers, and the hidden state in the layer $\ell \in (1, \cdots, L)$ is calculated as follows:

$$\mathbf{h}_t^{(\ell)} = \begin{cases} g^{(\ell)}(\mathbf{h}_t^{(\ell-1)}, \, \mathbf{h}_{t-1}^{(\ell)}) & (\ell = \text{odd}) \\ g^{(\ell)}(\mathbf{h}_t^{(\ell-1)}, \, \mathbf{h}_{t+1}^{(\ell)}) & (\ell = \text{even}) \end{cases} \qquad (3)$$

Both of the odd and even-numbered layers receive $\mathbf{h}_t^{(\ell-1)}$, the $t$-th hidden state of the $\ell - 1$ layer, as the first input of the function $g^{(\ell)}$, which is an arbitrary function [3]. As the second input of $g^{(\ell)}$, odd-numbered layers receive $\mathbf{h}_{t-1}^{(\ell)}$ while even-numbered layers receive $\mathbf{h}_{t+1}^{(\ell)}$. By calculating the hidden states until the $L$-th layer, we obtain a hidden state sequence $\mathbf{h}_{1:\mathrm{T}}^{(L)} = (\mathbf{h}_1^{(L)}, \cdots, \mathbf{h}_{\mathrm{T}}^{(L)})$. Using each vector $\mathbf{h}_t^{(L)}$, we calculate the probability of case labels for each word in the output layer.

### 3.3 Output Layer

In the output layer, multi-class classification is performed using the softmax function:

$$\mathbf{y}_t = \text{softmax}(\mathbf{W}_y \, \mathbf{h}_t^{(L)})$$

where $\mathbf{h}_t^{(L)}$ is a vector representation propagated from the last RNN layer (Fig 3). Each element of $\mathbf{y}_t$ is a probability value corresponding to each label. The label with the maximum probability among them is output as a result. In this task, there are five labels: NOM, ACC, DAT, PRED, null. The labels NOM, ACC and DAT indicate the nominative, accusative and dative case, respectively. PRED is the label for the predicate. null represents a word that does not play any case role.

---

[3]In this work, we use the Gated Recurrent Unit (GRU) (Cho et al., 2014) as the function $g^{(\ell)}$.

## 4 Multi-Sequence Model

While the single-sequence model assumes the independence between predicates, the multi-sequence model assumes the *multi-predicate interactions*. To capture such interactions between all predicates in a sentence, we extend the single-sequence model to the multi-sequence model using Grid-RNNs (Graves and Schmidhuber, 2009; Kalchbrenner et al., 2016). Figure 6 presents the overall architecture of the multi-sequence model, which consists of three components:

**Input Layer:** Map words to $M$ sequences of feature vectors for $M$ predicates.

**Grid Layer:** Update the hidden states over different sequences using Grid-RNNs.

**Output Layer:** Compute the probability of each case label for each word using the softmax function.

In the following subsections, we describe them in more detail.

### 4.1 Input Layer

The multi-sequence model takes as input a sentence $w_{1:\mathrm{T}} = (w_1, \cdots, w_{\mathrm{T}})$ and all predicates $\{p_m\}_1^M$ in the sentence. For each predicate $p_m$, the input layer creates a sequence of feature vectors $\mathbf{X}_m = (\mathbf{x}_{m,1}, \cdots, \mathbf{x}_{m,\mathrm{T}})$ by mapping each input word $w_t$ to a feature vector $\mathbf{x}_{m,t}$ based on Eq 1. That is, for $M$ predicates, $M$ sequences of feature vectors $\{\mathbf{X}_m\}_1^M$ are created.

Then, using Eq. 2, each feature vector $\mathbf{x}_{m,t}$ is mapped to $\mathbf{h}_{m,t}^{(0)}$, and a feature sequence is created for a predicate $p_m$, i.e., $\mathbf{H}_m^{(0)} = (\mathbf{h}_{m,1}^{(0)}, \cdots, \mathbf{h}_{m,\mathrm{T}}^{(0)})$. Consequently, for $M$ predicates, we obtain $M$ feature sequences $\{\mathbf{H}_m^{(0)}\}_1^M$.

### 4.2 Grid Layer

**Inter-Sequence Connections**

In the grid layers, we use Grid-RNNs to propagate the feature information over the different sequences (*inter-sequence connections*). The right figure in Figure 6 shows an odd-numbered layer of the Grid layers. The hidden state is recurrently calculated from upper-left ($m = 1, t = 1$) to lower-right ($m = M, t = \mathrm{T}$).

Formally, in the $\ell$-th layer, the hidden state $\mathbf{h}_{m,t}^{(\ell)}$ is calculated as follows:

$$\mathbf{h}_{m,t}^{(\ell)} = \begin{cases} g^{(\ell)}(\mathbf{h}_{m,t}^{(\ell-1)} \oplus \mathbf{h}_{m-1,t}^{(\ell)}, \, \mathbf{h}_{m,t-1}^{(\ell)}) & (\ell = \text{odd}) \\ g^{(\ell)}(\mathbf{h}_{m,t}^{(\ell-1)} \oplus \mathbf{h}_{m+1,t}^{(\ell)}, \, \mathbf{h}_{m,t+1}^{(\ell)}) & (\ell = \text{even}) \end{cases}$$

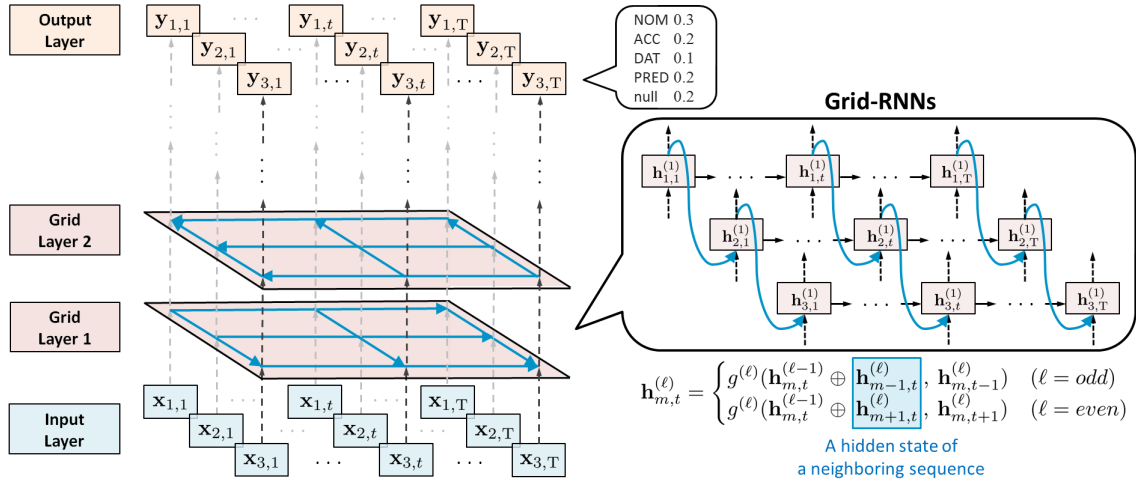

Figure 6: The overall architecture of the multi-sequence model: an example of three sequences.

This equation is similar to Eq. 3. The main difference is that the hidden state of a neighboring sequence, $\mathbf{h}_{m-1,t}^{(\ell)}$ (or $\mathbf{h}_{m+1,t}^{(\ell)}$), is concatenated ($\oplus$) with the hidden state of the previous ($\ell-1$) layer, $\mathbf{h}_{m,t}^{(\ell-1)}$, and is taken as input of the function $g^{(\ell)}$.

In the right figure in Figure 6, the blue curve lines represent the inter-sequence connections. Taking as input the hidden states of neighboring sequences, the network propagates feature information over multiple sequences (predicates). By calculating the hidden states until the $L$-th layer, we obtain $M$ sequences of the hidden states, i.e., $\{\mathbf{H}_m^{(L)}\}_1^M$, in which $\mathbf{H}_m^{(L)} = (\mathbf{h}_{m,1}^{(L)}, \cdots, \mathbf{h}_{m,\mathrm{T}}^{(L)})$.

**Residual Connections**

As more layers are stacked, it gets more difficult to learn the model parameters due to some problems such as gradient vanishment (Pascanu et al., 2013). In this work, we integrate residual connections (He et al., 2015; Wu et al., 2016) with our networks to connect between layers. Specifically, the input vector $\mathbf{h}_{m,t}^{(\ell-1)}$ of the $\ell$-th layer is added to the output vector $\mathbf{h}_{m,t}^{(\ell)}$. Residual connections can also be applied to the single-sequence model, and thus we perform the experiments on both models with/without residual connections.

### 4.3 Output Layer

Like the single-sequence model, using the softmax function, we calculate the probability of case labels of each word $w_t$ for each predicate $p_m$:

$$\mathbf{y}_{m,t} = \mathrm{softmax}(\mathbf{W}_y \, \mathbf{h}_{m,t}^{(L)})$$

where $\mathbf{h}_{m,t}^{(L)}$ is a hidden state vector calculated in the last Grid Layer.

## 5 Related Work

### 5.1 Japanese PAS Analysis Approaches

Existing approaches for Japanese PAS analysis are divided into two categories: (i) *pointwise approach* and (ii) *joint approach*. In the pointwise approach, we estimate the score of each argument candidate for one predicate, and select the argument candidate with the maximum score as an argument (Taira et al., 2008; Imamura et al., 2009; Hayashibe et al., 2011; Iida et al., 2016). In the joint approach, we select the score of all the predicate-argument combinations in one sentence, and select the combination with the highest score (Yoshikawa et al., 2011; Sasano and Kurohashi, 2011; Ouchi et al., 2015; Shibata et al., 2016). Compared with the pointwise approach, the joint approach has achieved better results.

### 5.2 Multi-Predicate Interactions

Ouchi et al. (2015) reported that it is beneficial to Japanese PAS analysis to capture the interactions between all predicates in a sentence. This is based on the linguistic intuition that the predicates in a sentence are semantically related to each other, and the interaction information can be a clue for PAS analysis.

Similarly, in semantic role labeling (SRL), Yang and Zong (2014) also reported that their reranking model capturing the multi-predicate interactions is effective for the English constituent-based SRL task (Carreras and Màrquez, 2005). Taking a step further in this direction, we propose the neural architecture that effectively models the multi-predicate interactions.

### 5.3 Neural Approaches

**Japanese PAS**

In recent years, several attempts have been made to apply neural networks for Japanese PAS analysis (Shibata et al., 2016; Iida et al., 2016)[4]. Shibata et al. (2016) used a feed-forward neural network for the score calculation part of the joint model proposed by Ouchi et al. (2015). Iida et al. (2016) used multi-column convolutional neural networks for the zero anaphora resolution task.

Both models exploited syntactic and selectional preference information as atomic features of neural networks. Using neural networks, the good performance was realized with mitigating the cost of manually designing combination features. In this work, we demonstrate that even without such syntactic information, our neural models realize the state-of-the-art performance by using word sequence information of a sentence.

**English SRL**

Some neural models achieved high performance without syntactic information in English SRL. Collobert et al. (2011) worked on the English constituent-based SRL task (Carreras and Màrquez, 2005). Their model exploited a convolutional neural network and achieved 74.15% in F-measure without syntactic information. Zhou and Xu (2015) also worked on the same task using bidirectional RNNs with CRF and achieved the state-of-the-art result, 81.07% in F-measure. Our models can be regarded as an extension from their model.

The main differences between Zhou and Xu (2015) and our work are: (i) the constituent vs dependency-based argument identification and (ii) the multi-predicate consideration. In the constituent-based SRL, since systems are required to identify the *spans* of arguments for each predicate, Zhou and Xu (2015) used CRF to capture the IOB label dependencies. In contrast, in Japanese dependency-based PAS analysis, since arguments are infrequently adjacent to each other, we replaced the CRF with the softmax function. Also, while the model of Zhou and Xu (2015) predicts arguments for each predicate independently, our multi-sequence model jointly predicts arguments for all predicates in a sentence at a time by considering the multi-predicate interactions.

---

[4]These previous studies used unpublished datasets and evaluated the performance with different experimental settings, so we cannot compare their models with ours.

## 6 Experiments

### 6.1 Experimental Settings

**Dataset**

We use the NAIST Text Corpus 1.5, which consists of 40,000 sentences of Japanese newspaper text (Iida et al., 2007). In the experiments, we adopt the standard data splits (Taira et al., 2008; Imamura et al., 2009; Ouchi et al., 2015):

**Train:** Articles: Jan 1-11,  Editorials: Jan-Aug
**Dev:**   Articles: Jan 12-13, Editorials: Sept
**Test:**  Articles: Jan 14-17, Editorials: Oct-Dec

We use the word boundaries annotated to the NAIST Text Corpus and the target predicates that have at least one argument in the same sentence. We do not use any external resources.

**Learning**

We train the model parameters by minimizing the cross-entropy loss function:

$$\mathcal{L}(\boldsymbol{\theta}) = -\sum_n \sum_t \log P(y_t|x_t) + \frac{\lambda}{2}||\boldsymbol{\theta}||^2$$

where $\theta$ is a set of model parameters, and the hyper-parameter $\lambda$ is the coefficient governing the L2 weight decay.

**Implementation Details**

We implement our neural models using Theano (Bastien et al., 2012). The number of epochs is set to 50, and we report the result of the test set in the epoch with the best F-measure of the development set. Parameter optimization is done by stochastic gradient descent method (SGD) using mini-batch, whose size is selected from $\{2, 4, 8\}$. The learning rate is automatically adjusted using Adam (Kingma and Ba, 2014). For the L2 weight decay, the hyper-parameter $\lambda$ in Eq. 4 is selected from $\{0.001, 0.0005, 0.0001\}$.

In the neural models, the number of the RNN and Grid layers are selected from $\{2, 4, 6, 8\}$. The window size $C$ for the PRED feature (Sec. 3.1) is set to $5$. Words with frequency $2$ or more are mapped to each word index, and the remaining words are mapped to the unknown word index. The dimensions $d_{word}$ and $d_{mark}$ of the embeddings are set to $32$. In the single-sequence model, the parameters of GRUs are set to $32 \times 32$. In the multi-sequence model, the parameters of GRUs related to the input values are set to $64 \times 32$, and

|  | Dep | Zero | All |
|---|---|---|---|
| Imamura+ 09 | 85.06 | 41.65 | 78.15 |
| Ouchi+ 15 | 86.07 | 44.09 | 79.23 |
| Single-Seq | 88.10 | 46.10 | 81.15 |
| Multi-Seq | **88.17** † | **47.12** † | **81.42** † |

Table 1: F-measures in the test set. Single-Seq is the single-sequence model, and Multi-Seq is the multi-sequence model. Imamura+ 09 is the model of Imamura et al. (2009) reimplemented by Ouchi et al. (2015), and Ouchi+ 15 is the ALL-Cases Joint Model of Ouchi et al. (2015). The mark † denotes the significantly better results with the significance level $p < 0.05$ comparing between Single-Seq and Multi-Seq.

others are $32 \times 32$. The initial values of all the parameters are sampled according to the uniform distribution from $[-\frac{\sqrt{6}}{\sqrt{row+col}}, \frac{\sqrt{6}}{\sqrt{row+col}}]$, where $row$ and $col$ are the number of rows and columns of each matrix, respectively.

**Baseline Models**
We compare our models with the models in the previous works (Sec. 5.1) that use the NAIST Text Corpus 1.5. As a baseline of the pointwise approach, we use the pointwise model[5] of Imamura et al. (2009). In addition, as a baseline of the joint approach, we use the model of Ouchi et al. (2015), which achieved the best result on the NAIST Text Corpus 1.5.

### 6.2 Results

**Neural Models vs Baseline Models**
Table 1 presents F-measures of our neural sequence models with 8 RNN or Grid layers and the baseline models on the test set, in which as the significant test, we used the bootstrap resampling method. In all the metrics, both of the single-sequence (Single-Seq) and multi-sequence model (Multi-Seq) outperformed the baseline models. This confirms that our neural sequence models realize high-performance even without syntactic information by learning contextual information effective for PAS analysis from a word sequence of the sentence.

In particular, for zero arguments (Zero), our models achieved a considerable improvement compared with the state-of-the-art model of

---

[5]We compared the results of the model reimplemented by Ouchi et al. (2015).

| L |  | Single-Seq +res. | Single-Seq −res. | Multi-Seq +res. | Multi-Seq −res. |
|---|---|---|---|---|---|
| 2 | Dep | **87.34** | 87.10 | 87.43 | **87.73** |
|   | Zero | **47.98** | 47.90 | **47.66** | 46.93 |
|   | All | **80.62** | 80.24 | **80.71** | 80.68 |
| 4 | Dep | 87.27 | **87.41** | **87.60** | 87.09 |
|   | Zero | 50.43 | **50.83** | 48.10 | **48.58** |
|   | All | 80.92 | **80.99** | **80.99** | 80.59 |
| 6 | Dep | **87.73** | 87.11 | **88.04** | 87.39 |
|   | Zero | 48.81 | **49.51** | **48.98** | 48.91 |
|   | All | **81.05** | 80.63 | **81.19** | 80.68 |
| 8 | Dep | **87.98** | 87.23 | **87.65** | 87.07 |
|   | Zero | 47.40 | **48.38** | **49.34** | 48.23 |
|   | All | **81.31** | 80.33 | **81.33** | 80.40 |

Table 2: Performance comparison for different numbers of layers on the development set in F-measures. $L$ is the number of the RNN or Grid layers. $+res.$ or $-res.$ indicates whether the model has the residual connections ($+$) or not ($-$).

Ouchi et al. (2015), i.e., the single model improved around 2.0 points and the multi-sequence model improved around 3.0 points in F-measure. These results suggest that it is beneficial to Japanese PAS analysis, particularly to the zero argument identification, to model the context in the entire sentence using RNNs.

**Effects of Multiple Predicate Consideration**
As Table 1 shows, the multi-sequence model significantly outperformed the single-sequence model in F-measure in total (81.42% vs 81.15%). This result demonstrates that the grid-type neural architecture can effectively capture the multi-predicate interactions by connecting between the sequences of the argument candidates for all predicates in a sentence.

Compared with the single-sequence model for different argument types, the multi-sequence model achieved slightly but significantly better result for the direct dependency arguments (Dep) (88.10% vs 88.17%). In addition, for zero arguments (Zero), which have no syntactic dependency with its predicate, the multi-sequence model significantly outperformed the single-sequence model by around 1.0 points in F-measure (46.10% vs 47.12%). This shows that capturing the multi-predicate interactions is particularly effective for zero arguments, which is consistent with the results of Ouchi et al. (2015).

|  | Dep | | | Zero | | |
|---|---|---|---|---|---|---|
|  | NOM | ACC | DAT | NOM | ACC | DAT |
| NAIST Text Corpus 1.5 | | | | | | |
| Imamura+ 09 | 86.50 | 92.84 | 30.97 | 45.56 | 21.38 | 0.83 |
| Ouchi+ 15 | 88.13 | 92.74 | 38.39 | 48.11 | 24.43 | 4.80 |
| Single-Seq | 88.32 | 93.89 | 65.91 | 49.51 | 35.07 | 9.83 |
| Multi-Seq | 88.75 | 93.68 | 64.38 | 50.65 | 32.35 | 7.52 |
| NAIST Text Corpus 1.4$\beta$ | | | | | | |
| Taira+ 08* | 75.53 | 88.20 | 89.51 | 30.15 | 11.41 | 3.66 |
| Imamura+ 09* | 87.0 | 93.9 | 80.8 | 50.0 | 30.8 | 0.0 |
| Sasano+ 11* | - | - | - | 39.5 | 17.5 | 8.9 |

Table 3: Performance comparison for different case roles on the test set in F-measures. NOM, ACC or DAT is the nominal, accusative or dative case, respectively. The mark * indicates that the model uses external resources.

**Effects of Network Depth**

Table 2 presents F-measures of the neural sequence models with different network depths and with/without residual connections. The performance tends to get better as the RNN or Grid layers get deeper with residual connections. In particular, the two models with 8 layers and residual connections achieved considerable improvements of around 1.0 point in F-measure compared the models without residual connections, which means that the residual connections contribute to the effective parameter learning of deeper models.

**Comparison per Case Role**

Table 3 shows F-measures for each case role. For reference, we show the results of the previous studies using NAIST Text Corpus 1.4$\beta$ with external resources as well.[6]

Comparing between the models using the NAIST Text Corpus 1.5, the single-sequence and multi-sequence models outperformed the baseline models in all the metrics. In particular, for the dative case, the two neural models achieved much higher results by around 30 points. This suggests that although dative arguments appear infrequently compared with the other two case arguments, the neural models can robustly learn it.

In addition, for zero arguments (*Zero*), the neural models achieved better results than the baseline models. Especially, for zero arguments of

---

[6]The major difference between NAIST Text Corpus 1.4$\beta$ and 1.5 is the revision of the annotation criterion for the dative case (DAT) (corresponding to Japanese case marker "に"). Argument and adjunct usages of the case marker "に" are not distinguished in 1.4$\beta$, making the identification of the dative case seemingly easy (Ouchi et al., 2015).

the nominative case (NOM), the multi-sequence model achieved a considerable improvement of around 2.5 points in F-measure compared with the state-of-the-art model of Ouchi et al. (2015). To achieve high accuracies for the analysis of such zero arguments, it is necessary to capture long distance dependencies (Iida et al., 2005; Sasano and Kurohashi, 2011; Iida et al., 2015). Therefore, the improvements of the results suggest that the neural models effectively capture long distance dependencies using RNNs that can encode the context in the entire sentence.

## 7 Conclusion

In this work, we introduced neural sequence models that automatically induce effective feature representations from word sequence information of a sentence for Japanese PAS analysis. The experiments on NAIST Text Corpus 1.5 demonstrated that the models achieve the state-of-the-art result without syntactic information. In particular, our multi-sequence model improved the performance for *zero argument* identification, one of the problematic issues in Japanese PAS analysis, by considering the *multi-predicate interactions* using Grid-RNNs.

Since our neural models are applicable to SRL, applying our models for multilingual SRL tasks is an interesting line of the future research. In addition, in this work, the model parameters were learned without any external resources. For future work, we plan to explore effective methods for exploiting large-scale unlabeled data to learn the neural models.

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
