# Peer review of "Neural Modeling of Multi-Predicate Interactions for Japanese Predicate Argument Structure Analysis"

_ACL 2017 — decision unknown_

[Official Review · Reviewer 1 · rating 4 · confidence 3]
soundness 5 · originality 5 · clarity 4 · impact 3 · substance 4 · appropriateness 5 · meaningful comparison 3 · presentation format Oral Presentation

- Strengths:

This paper presents a sophisticated application of Grid-type Recurrent Neural
Nets to the task of determining predicate-argument structures (PAS) in
Japanese.  The approach does not use any explicit syntactic structure, and
outperforms the current SOA systems that do include syntactic structure.  The
authors give a clear and detailed description of the implementation and of the
results.  In particular, they pay close attention to the performance on dropped
arguments, zero pronouns, which are prevalent in Japanese and especially
challenging with respect to PAS. Their multi-sequence model, which takes all of
the predicates in the sentence into account, achieves the best performance for
these examples.  The paper is detailed and clearly written.

- Weaknesses:

I really only have minor comments. There are some typos listed below, the
correction of which would improve English fluency. I think it would be worth
illustrating the point about the PRED including context around the "predicate"
with the example from Fig 6 where the accusative marker is included with the
verb in the PRED string.  I didn't understand the use of boldface in Table 2,
p. 7.

- General Discussion:

Typos:

p1 :  error propagation does not need a "the", nor does "multi-predicate
interactions"
p2: As an solution -> As a solution, single-sequence model -> a single-sequence
model,                    multi-sequence model -> a multi-sequence model 
p. 3 Example in Fig 4.                    She ate a bread -> She ate bread.
p. 4 assumes the independence -> assumed independence, the multi-predicate
interactions -> multi-predicate interactions, the multi-sequence model -> a
multi-sequence model
p.7: the residual connections -> residual connections, the multi-predicate
interactions -> multi-predicate interactions (twice)
p8 NAIST Text Corpus -> the NAIST Text Corpus, the state-of-the-art result ->
state-of-the-art results

I have read the author response and am satisfied with it.

[Official Review · Reviewer 2 · rating 4 · confidence 5]
soundness 5 · originality 5 · clarity 4 · impact 3 · substance 4 · appropriateness 5 · meaningful comparison 3 · presentation format Oral Presentation

This paper proposes new prediction models for Japanese SRL task by adopting the
English state-of-the-art model of (Zhou and Xu, 2015).
The authors also extend the model by applying the framework of Grid-RNNs in
order to handle the interactions between the arguments of multiple predicates.

The evaluation is performed on the well-known benchmark dataset in Japanese
SRL, and obtained a significantly better performance than the current state of
the art system.

Strengths:
The paper is well-structured and well-motivated.
The proposed model obtains an improvement in accuracy compared with the current
state of the art system.
Also, the model using Grid-RNNs achieves a slightly better performance than
that of proposed single-sequential model, mainly due to the improvement on the
detection of zero arguments, that is the focus of this paper.

Weakness:
To the best of my understanding, the main contribution of this paper is an
extension of the single-sequential model to the multi-sequential model. The
impact of predicate interactions is a bit smaller than that of (Ouchi et al.,
2015). There is a previous work (Shibata et al., 2016) that extends the (Ouchi
et al., 2015)'s model
with neural network modeling. I am curious about the comparison between them.

[Official Review · Reviewer 3 · rating 4 · confidence 4]
soundness 5 · originality 5 · clarity 4 · impact 3 · substance 4 · appropriateness 5 · meaningful comparison 3 · presentation format Poster

This paper proposes a joint neural modelling approach to PAS analysis in
Japanese, based on Grid-RNNs, which it compares variously with a conventional
single-sequence RNN approach.

This is a solidly-executed paper, targeting a well-established task from
Japanese but achieving state-of-the-art results at the task, and presenting
the task in a mostly accessible manner for those not versed in
Japanese. Having said that, I felt you could have talked up the complexity of
the task a bit, e.g. wrt your example in Figure 1, talking through the
inherent ambiguity between the NOM and ACC arguments of the first predicate,
as the NOM argument of the second predicate, and better describing how the
task contrasts with SRL (largely through the ambiguity in zero pronouns). I
would also have liked to have seen some stats re the proportion of zero
pronouns which are actually intra-sententially resolvable, as this further
complicates the task as defined (i.e. needing to implicitly distinguish
between intra- and inter-sentential zero anaphors). One thing I wasn't sure of
here: in the case of an inter-sentential zero pronoun for the argument of a
given predicate, what representation do you use? Is there simply no marking of
that argument at all, or is it marked as an empty argument? My reading of the
paper is that it is the former, in which case there is no explicit
representation of the fact that there is a zero pronoun, which seems like a
slightly defective representation (which potentially impacts on the ability of
the model to capture zero pronouns); some discussion of this would have been
appreciated.

There are some constraints that don't seem to be captured in the model (which
some of the ILP-based methods for SRL explicitly model, e.g.): (1) a given
predicate will generally have only one argument of a given type (esp. NOM and
ACC); and (2) a given argument generally only fills one argument slot for a
given predicate. I would have liked to have seen some analysis of the output
of the model to see how well the model was able to learn these sorts of
constraints. More generally, given the mix of numbers in Table 3 between
Single-Seq and Multi-Seq (where it is really only NOM where there is any
improvement for Multi-Seq), I would have liked to have seen some discussion of
the relative differences in the outputs of the two models: are they largely
identical, or very different but about the same in aggregate, e.g.? In what
contexts do you observe differences between the two models? Some analysis like
this to shed light on the internals of the models would have made the
difference between a solid and a strong paper, and is the main area where I
believe the paper could be improved (other than including results for SRL, but
that would take quite a bit more work).

The presentation of the paper was good, with the Figures aiding understanding
of the model. There were some low-level language issues, but nothing major:

l19: the error propagation -> error propagation
l190: an solution -> a solution
l264 (and Figure 2): a bread -> bread
l351: the independence -> independence
l512: the good -> good
l531: from their model -> of their model
l637: significent -> significance
l638: both of -> both

and watch casing in your references (e.g. "japanese", "lstm", "conll", "ilp")